# Sensitivity of Inner Spacer Thickness Variations for Sub-3-nm Node Silicon Nanosheet Field-Effect Transistors

**DOI:** 10.3390/nano12193349

**Published:** 2022-09-26

**Authors:** Sanguk Lee, Jinsu Jeong, Jun-Sik Yoon, Seunghwan Lee, Junjong Lee, Jaewan Lim, Rock-Hyun Baek

**Affiliations:** Department of Electrical Engineering, Pohang University of Science and Technology (POSTECH), Pohang 37673, Gyeongbuk, Korea

**Keywords:** nanosheet FET, inner spacer, inner spacer thickness variation, performance sensitivity, source/drain recess depth, TCAD simulation

## Abstract

The inner spacer thickness (T_IS_) variations in sub-3-nm, node 3-stacked, nanosheet field-effect transistors (NSFETs) were investigated using computer-aided design simulation technology. Inner spacer formation requires a high selectivity of SiGe to Si, which causes inevitable T_IS_ variation (ΔT_IS_). The gate length (L_G_) depends on the T_IS_. Thus, the DC/AC performance is significantly affected by ΔT_IS_. Because the effects of ΔT_IS_ on the performance depend on which inner spacer is varied, the sensitivities of the performance to the top, middle, and bottom (T, M, and B, respectively) ΔT_IS_ should be studied separately. In addition, the source/drain (S/D) recess process variation that forms the parasitic bottom transistor (tr_pbt_) should be considered with ΔT_IS_ because the gate controllability over tr_pbt_ is significantly dependent on ΔT_IS,B_. If the S/D recess depth (T_SD_) variation cannot be completely eliminated, reducing ΔT_IS,B_ is crucial for suppressing the effects of tr_pbt_. It is noteworthy that reducing ΔT_IS,B_ is the most important factor when the T_SD_ variation occurs, whereas reducing ΔT_IS,T_ and ΔT_IS,M_ is crucial in the absence of T_SD_ variation to minimize the DC performance variation. As the T_IS_ increases, the gate capacitance (C_gg_) decreases owing to the reduction in both parasitic and intrinsic capacitance, but the sensitivity of C_gg_ to each ΔT_IS_ is almost the same. Therefore, the difference in performance sensitivity related to AC response is also strongly affected by the DC characteristics. In particular, since T_SD_ of 5 nm increases the off-state current (I_off_) sensitivity to ΔT_IS,B_ by a factor of 22.5 in NFETs, the ΔT_IS,B_ below 1 nm is essential for further scaling and yield enhancement.

## 1. Introduction

Silicon fin-shaped field-effect transistors (FinFETs) have been continuously scaled down from 22-nm to 5-nm nodes using fins with high aspect ratios and design technology co-optimization [1,2,3,4,5,6]. However, increasing the fin aspect ratio is challenging owing to the process complexity, and FinFETs with narrow fins exhibit threshold-voltage variations and performance degradation induced by the quantum confinement effect [7,8,9,10]. By contrast, Silicon gate-all-around nanosheet field-effect transistors (NSFETs) have received considerable attention as promising devices that can replace FinFETs in sub-3-nm nodes, as they can overcome these limitations through stacked nanosheet (NS) channels [11]. Furthermore, NSFETs provide excellent electrostatics because the gate surrounds the NS channels and drives a larger current within the same footprint with a wider effective channel width than FinFETs [11,12].

The inner spacer is a distinctive structural feature of NSFETs that has not been employed in previous devices. Typically, selective etching of the SiGe sacrificial layers is performed to form the inner spacer. However, selective etching requires a high selectivity of SiGe to Si and lateral etching. Therefore, it can be vulnerable to process variations [13,14]. Furthermore, because the inner spacer determines the gate length (L_G_), these variations result in NSFETs with unintended L_G_ changes and cause unoptimized leakage and DC/AC performance [11,15]. Therefore, precise control of the inner spacer thickness (T_IS_) is crucial for performance optimization.

Previous studies related to the inner spacer have focused on the electrical properties of NSFETs, assuming the same shape and thickness from the top inner spacer to the bottom inner spacer [11,15]. However, in the actual process, the T_IS_ variation (ΔT_IS_) may not occur uniformly [11,16]. In addition, for three-stacked NSFETs, the top and middle inner spacers adjoin two adjacent NS channels, while the bottom inner spacer adjoins only one NS channel and a punch-through stopper (PTS) region. Thus, the thickness variations of the top/middle/bottom (T/M/B) inner spacers have different effects on the device behavior; i.e., the T/M/B ΔT_IS_ (ΔT_IS,T_/ΔT_IS,M_/ΔT_IS,B_) have different effects on the performance. Therefore, the performance sensitivities must be studied separately. Additionally, the over-etched S/D recess is a crucial factor determining the effects of the parasitic bottom transistor (tr_pbt_) on the DC performance [17]. The effects of tr_pbt_ on performance become more pronounced as L_G_ decreases, which is a potential threat for further scaling [17,18]. However, there have been no studies on the effects of the S/D recess depth (T_SD_) along with T/M/B ΔT_IS_ on the device behavior. In this study, for the first time, we comprehensively analyzed the sensitivity of the DC/AC characteristics to each ΔT_IS_ considering the T_SD_, and the off-state characteristics were analyzed in detail using fully calibrated computer-aided design (TCAD) simulation technology [19].

## 2. Device Structure and Simulation Methodology

The sub-3-nm node NSFETs investigated in this study were simulated using Sentaurus TCAD tools. The following physical models were considered in the TCAD simulation: The drift–diffusion model was considered using Poisson’s equations and the continuity equations to determine the electrostatic potential and carrier transport.The density gradient model was considered for the quantum confinement effect in the drift-diffusion model [20,21].The Slotboom bandgap narrowing model was considered for doping-dependent bandgap narrowing in Si and SiGe [22,23].A low-field ballistic mobility model was considered for quasi-ballistic transport [24].Mobility degradation at the interfaces was considered for remote phonon scattering and remote Coulomb scattering [25].The inversion and accumulation layer mobility models were considered for Coulomb impurity, phonon scattering, and surface roughness scattering [26].A high-field saturation model was considered for carrier velocity saturation under a strong electric field [27].The deformation potential model was considered for the strain-induced density of states, effective mass of carriers, and energy-band shift [28].The Auger and Shockley–Read–Hall (SRH) recombination models were used.

Figure 1a shows schematics of the sub-3-nm node 3-stacked NSFETs. Among the T/M/B ΔT_IS_, we varied only one of the T/M/B T_IS_, with the others fixed at 5 nm, to investigate the effects of the T/M/B ΔT_IS_ on the DC/AC characteristics separately. Here, the thicknesses of the T/M/B inner spacers were defined as T_IS,T_, T_IS,M_, and T_IS,B_, respectively. In addition, T_SD_ of 0 and 5 nm were used to consider the effects of T_SD_ on the performance along with those of ΔT_IS_ [14]. Therefore, a comprehensive analysis of ΔT_IS_ considering the T_SD_ effect was performed.

The T_IS_ without variation (T_IS,ref_) was set as 5 nm, and only one of the three T_IS_ was varied from 3 to 7 nm (Figure 1b). In this study, ΔT_IS_ was defined as T_IS_ − T_IS,ref_, and the L_G_ of each channel depended on ΔT_IS_ (L_G_ = 22 − 2 × (T_IS,ref_ + ΔT_IS_)). Si_0.98_C_0.02_ (Si_0.5_Ge_0.5_) S/D doped with phosphorus (boron) at 4 × 10^20^ cm^−3^ was used for the NFETs (PFETs). The contact resistance of the S/D was set as 1 nΩ·cm^2^. The PTS layer was doped at 3 × 10^18^ cm^−3^, and the drain voltage (V_ds_) was fixed at |0.7| V. The geometric parameters are presented in Table 1. The NSFETs were calibrated to TSMC’s 5-nm node FinFETs [5], and the same physical parameters were used, as shown in our previous studies [29]. The drain current was fitted by adjusting the doping profile, ballistic coefficient, and saturation velocity. The doping profile was changed to fit the subthreshold swing and DIBL since the doping profile is deeply concerned with the device behaviors in the subthreshold region. The ballistic coefficient was tuned to fit the drain current in the linear region, and the saturation velocity was set to fit the drain current in the saturation region. We extracted the on-state current (I_on_) and gate capacitance (C_gg_) at |V_gs_| = 0.7 V and |V_ds_| = 0.7 V. Moreover, the off-state current (I_off_) and parasitic capacitance (C_para_) were extracted at |V_gs_| = 0 V and |V_ds_| = 0.7 V.

## 3. Results and Discussion

Figure 2 shows the transfer curves of NSFETs with different T_IS,B_ for T_SD_ = 0 and 5 nm. No significant dependence of the DC performance on ΔT_IS,B_ was observed at T_SD_ = 0 (Figure 2a). By contrast, at T_SD_ = 5 nm, the I_off_ increased significantly as T_IS,B_ increased (Figure 2b). The T_SD_ typically impacts the I_off_ of tr_pbt_ [17], where T_IS,B_ determines the L_G_ of tr_pbt_. Because the L_G_ of tr_pbt_ affects the gate controllability over the PTS region, an increase in ΔT_IS,B_ significantly degrades the DC performance. As an increase in T_SD_ degrades the gate controllability of tr_pbt_, T_IS,B_ is a critical factor determining the parasitic punch-through current (I_pt_) in the PTS region. Therefore, the subthreshold swing and DIBL are significantly degraded, as shown in the inset of Figure 2 and Table 2. 

The I_off_ sensitivities to the T/M/B ΔT_IS_ (S_Ioff,T_/S_Ioff,M_/S_Ioff,B_) are compared in Figure 3. We defined S_Ioff_ as the slope of I_off_−ΔT_IS_, which indicates how sensitively I_off_ varies with respect to ΔT_IS_. For the NFETs with T_SD_ = 0 nm, the S_Ioff,T_ (0.208) and S_Ioff,M_ (0.228) slightly exceeded the S_Ioff,B_ (0.104 nA/nm), and similar S_Ioff_ tendencies were observed for the PFETs. The T_SD_ variation not only increased I_off_, but also significantly increased S_Ioff,B_ for both the NFETs and the PFETs. The S_Ioff,B_ for the NFETs is greater than that for the PFETs, which is mainly attributed to the S/D dopant diffusion into the PTS region. Phosphorus has a higher diffusivity than boron; therefore, more S/D dopant diffuses into the PTS region in NFETs than in PFETs [30]. Consequently, the NFETs are more sensitive to the ΔT_IS,B_ in terms of I_off_. For the NFETs with T_SD_ = 5 nm, S_Ioff,T_, S_Ioff,M_, and S_Ioff,B_ were 0.195, 0.209, and 2.34 nA/nm, respectively. S_Ioff,T_ and S_Ioff,M_ were almost identical regardless of the T_SD_, but S_Ioff,B_ increased by a factor of 22.5 when the T_SD_ increased from 0 to 5 nm. This indicated that the S/D recess process variation slightly affects S_Ioff,T_ and S_Ioff,M_ but significantly affects S_Ioff,B_. Thus, if the T_SD_ variation is not perfectly eliminated, ΔT_IS,B_ should be controlled below 1 nm, because devices with greater than 10 times in I_off_ are not suitable for the intended system-on-chip applications.

The differences in the S_Ioff_ shown in Figure 3 can be explained using the I_off_-density profiles (Figure 4). In NSFETs with T_SD_ = 0 nm, most carriers existed in the NS channels, and a few were in the PTS region owing to the heavily doped PTS. Furthermore, ΔT_IS_-induced I_off_ density variations mainly arose in the NS channels next to the inner spacer with variations in the thickness. Thus, the top and middle inner spacers adjacent to the NS channels with high carrier concentrations exhibited larger changes in the I_off_ density than the bottom inner spacer. Therefore, S_Ioff,T_ and S_Ioff,M_ are higher than S_Ioff,B_ for the NSFETs with T_SD_ = 0 nm. By contrast, S_Ioff,B_ was the highest when the T_SD_ was 5 nm. Figure 4b shows the I_off_ density profiles for NFETs with different T_IS,B_ in the case of T_SD_ = 5 nm. As T_IS,B_ increased, the off-state I_pt_ (I_pt,off_) was not suppressed, resulting in a significant increase in I_off_, as shown in Figure 2. The I_off_ density varied according to ΔT_IS,B_ in the bottom NS and PTS regions but varied to a significantly larger extent in the PTS region. Specifically, the T_SD_ variation significantly enhanced the effects of tr_pbt_ on I_off_, and the change in I_pt,off_ was a dominant factor in the S_Ioff,B_ increment. This is because the PTS region was only controlled by the bottom gate. Therefore, the bottom gate could not effectively control the PTS region far from the bottom gate. As a result, worse short-channel effects (SCEs) were observed in the PTS region than in the NS channel.

Figure 5a shows the conduction band energy (E_c_) diagrams of the source–PTS–drain in the NFETs, which were extracted under the off-state bias condition. As the T_SD_ increased from 0 to 5 nm, the significant reduction in the energy barrier height (Φ_b_) from 478 to 402 mV was caused by the larger amount of S/D dopant diffusion into the PTS region at a T_SD_ of 5 nm. In NFETs with T_SD_ = 0 nm, the Φ_b_ of the PTS region was sufficiently high to control I_pt,off_ regardless of ΔT_IS,B_ (Figure 5b). Therefore, I_off_ can be effectively controlled even with ΔT_IS,B_. However, if Φ_b_ is not sufficiently high, the additional Φ_b_ reduction due to ΔT_IS,B_ can be a critical factor in inducing I_pt,off_. An additional Φ_b_ reduction was observed when T_IS,B_ increased, and the change in Φ_b_ by ΔT_IS,B_ significantly contributed to the I_pt,off_ variation (Figure 3 and Figure 5c). Therefore, the bottom L_G_ of tr_pbt_, which is related to T_IS,B_, is important for suppressing SCEs in the PTS region. According to these results, S_Ioff,B_ is significantly affected by T_SD_. Thus, minimizing ΔT_IS,B_ is more crucial when an over-etched S/D recess occurs.

Figure 6 shows the relationship between the on-state current (I_on_) and ΔT_IS_, and the slope indicates the I_on_ sensitivity (S_Ion_). For the NFETs, the S_Ion,T_ and S_Ion,M_ are slightly higher than the S_Ion,B_ regardless of the T_SD_. By contrast, for the PFETs, the S_Ion,B_ varied significantly with respect to the T_SD_, leading to an increase in S_Ion,B_ by a factor of 1.9. Thus, an increase in ΔT_IS,B_ can cause severe I_on_ variations when the T_SD_ is not precisely controlled. The reason for the differences in the S_Ion_ is explained in Figure 7.

The R_sd_ sensitivity (S_Rsd_) and on-state I_pt_ (I_pt,on_)-density variations to the ΔT_IS_ account for the differences in T/M/B S_Ion_ (Figure 7). R_sd_ was extracted using Y-function techniques, as described in [31]. Two main factors determine S_Ion_: R_sd_ and inversion charges in the PTS region. Additionally, the major factors affecting S_Ion_ depend on the T_SD_. For both the NFETs and PFETs with T_SD_ = 0 nm, S_Ion_ was mainly affected by the change in R_sd_, which consisted of the series S/D epi resistance (R_epi_) and extension resistance (R_ext_). R_epi_ did not change with respect to ΔT_IS_, but R_ext_ did. Because S_Rsd_ varied proportionally to the number of NS channels adjacent to the inner spacer where ΔT_IS_ occurred (Figure 7a), S_Ion,T_ and S_Ion,M_ were greater than S_Ion,B_. However, the inversion charges in the PTS region significantly affected S_Ion_ when T_SD_ was 5 nm. As the deep T_SD_ caused a substantial current to flow through tr_pbt_, the I_on_ contribution of the PTS region was no longer small. The inversion charges in the PTS region should also be considered (Figure 7b). For the NFETs, the I_pt,on_ density in tr_pbt_ decreased slightly as T_IS,B_ increased, whereas the large decrease in I_pt,on_ was observed for the PFETs. This is because higher SCEs and V_th_ reductions were observed in the NFETs, as the large amounts of diffused S/D dopants reduced Φ_b_ (Figure 2b and Figure 5a). Therefore, in the NFETs, the V_th_ reduction of tr_pbt_ lowered the effects of the increase in R_sd_, which was the dominant factor determining S_Ion,B_. By contrast, in the PFETs, the V_th_ reduction of tr_pbt_ was small; thus, I_pt,on_ decreased significantly owing to the increase in the R_sd_ of tr_pbt_. Consequently, S_Ion,B_ was the smallest for the NFETs, but for the PFETs, the T_SD_ variation caused I_on_ to be most sensitive to ΔT_IS,B_.

Based on these results, we can provide two guidelines for controlling the DC performance variation, which depends on T_SD_. In the case of T_SD_ = 0, precisely controlling T_IS,T_ and T_IS,M_ rather than T_IS,B_ is effective for minimizing the variations in I_off_ and I_on_, as shown in Figure 3 and Figure 6. However, considering the T_SD_ variation, it is necessary to focus on the bottom inner spacer, because a precisely controlled T_IS,B_, can considerably reduce the performance variation. Otherwise, the effects of tr_pbt_ on the DC performance become large as T_IS,B_ increases, resulting in the worst case with the highest I_off_ and lowest I_on_ in PFETs, which significantly diminishes the performance advantages of NSFETs.

The gate capacitance (C_gg_) with respect to ΔT_IS_ for NSFETs (T_SD_ = 0) is shown in Figure 8, and C_gg_ is decomposed into the intrinsic capacitance (C_int_) and parasitic capacitance (C_para_). C_para_ was extracted under the off-state bias, and C_int_ was calculated by subtracting C_para_ from C_gg_ under the on-state bias. As shown in Figure 8a, the differences in the C_gg_ sensitivity to T/M/B ΔT_IS_ (S_Cgg_) were small. However, the changes in C_int_ and C_para_ for each ΔT_IS_ did not have the same sensitivity. C_para_, which was determined by the fringing field between the gate and S/D, was affected by the T_IS_. Therefore, the sensitivity of C_para_ to ΔT_IS_ was almost identical among the T/M/B ΔT_IS_ (Figure 8b). However, the sensitivity of C_int_ to ΔT_IS,B_ was lower than those of ΔT_IS,T_ and ΔT_IS,M_ (Figure 8c). Although the inversion charge variations caused by ΔT_IS,B_ mainly occurred in the bottom NS and PTS regions, the charge variations in the PTS region were smaller than those in the NS channels, leading to different AC sensitivities to the T/M/B ΔT_IS_. However, because the differences in the C_int_ sensitivity to the T/M/B ΔT_IS_ were not large, it can be concluded that the overall performance sensitivity difference induced by each ΔT_IS_ has greater effects on DC (I_off_, I_on_) rather than the AC performance.

## 4. Conclusions

The sensitivities of the DC/AC performance to the T/M/B ΔT_IS_ in sub-3-nm node NSFETs were quantitatively investigated using a fully calibrated TCAD simulation. The DC performance sensitivities (I_off_, I_on_) to the T/M/B ΔT_IS_ differed. However, there were no significant differences in the AC sensitivities. One of the notable results was that ΔT_IS_, which varied the performance the most, was different according to the T_SD_ variations. In NSFETs with T_SD_ = 0 nm, S_Ioff,B_ was lower than S_Ioff,T_ and S_Ioff,M_ because the effects of ΔT_IS,B_ were primarily observed in the bottom NS channel. However, tr_pbt_ was no longer negligible when the T_SD_ was 5 nm. Thus, if the T_SD_ variation is not controlled, NFETs (PFETs) have higher S_Ioff,B_ (S_Ion,B_) because of the effects of tr_pbt_. It can be concluded that the bottom inner spacer is the element with the most significant effect on the DC/AC performance. Hence, reducing ΔT_IS,B_ is important for yield enhancement.

## Figures and Tables

**Figure 1 nanomaterials-12-03349-f001:**
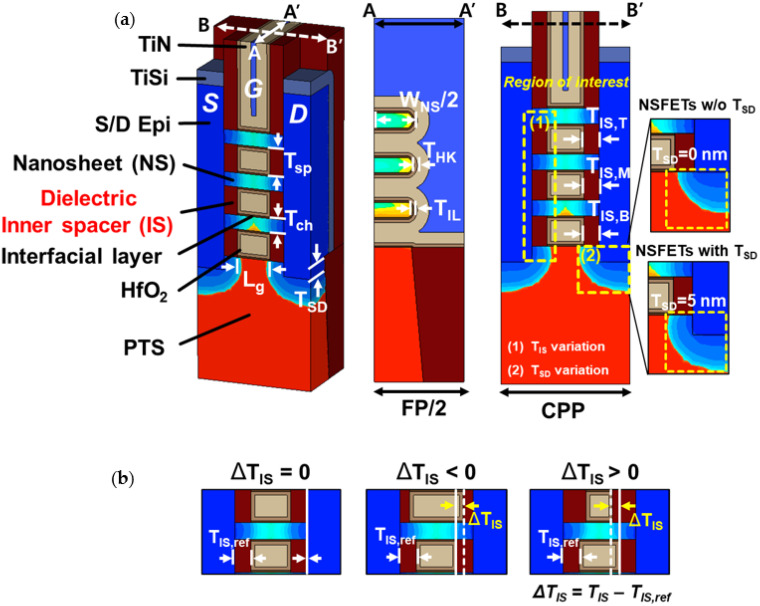
(**a**) Structure of NSFETs with the T_SD_ and cross-sectional views. (**b**) Schematics of ΔT_IS_ and its definition.

**Figure 2 nanomaterials-12-03349-f002:**
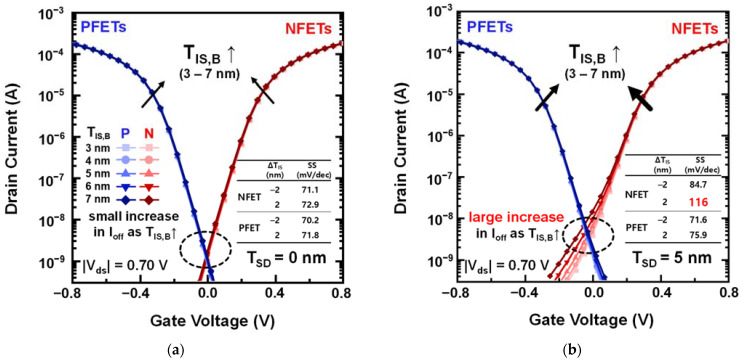
Transfer curves of the NSFETs having different T_IS,B_ with (**a**) T_SD_ = 0 nm and (**b**) T_SD_ = 5 nm.

**Figure 3 nanomaterials-12-03349-f003:**
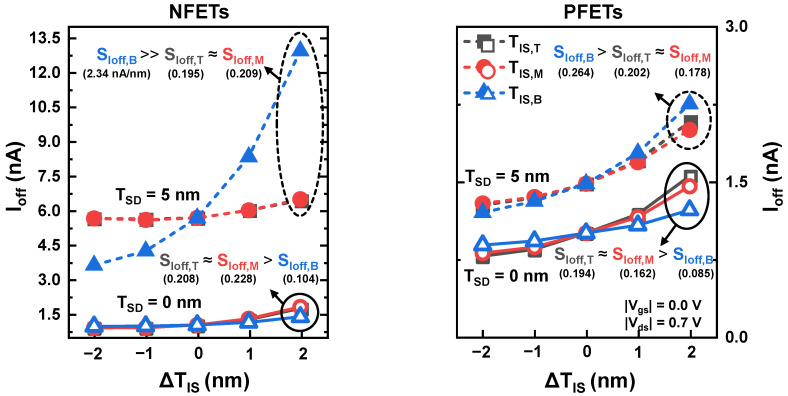
I_off_ of NSFETs according to ΔT_IS_ with T_SD_ = 0 and 5 nm.

**Figure 4 nanomaterials-12-03349-f004:**
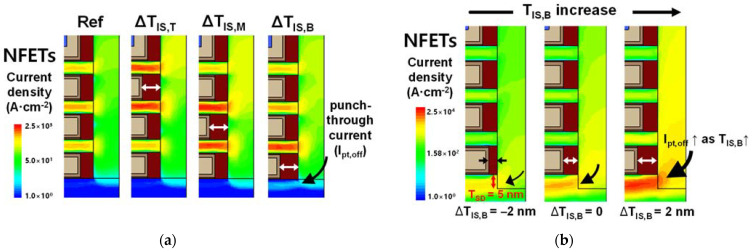
(**a**) I_off_ density profiles of the NFETs with T_SD_ = 0 and each ΔT_IS_ equal to 2 nm. (**b**) I_off_ density profiles of the NFETs with T_SD_ = 5 nm for different values of ΔT_IS,B_.

**Figure 5 nanomaterials-12-03349-f005:**
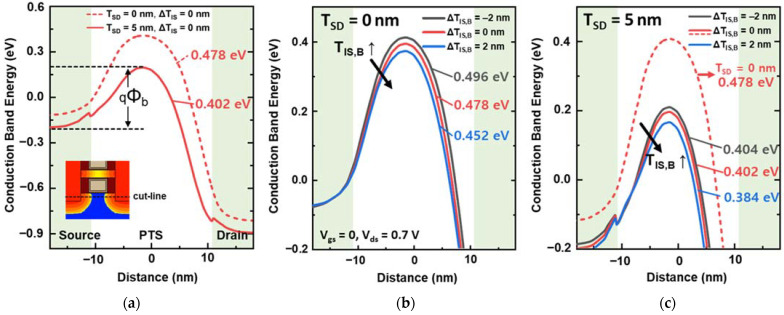
(**a**) Energy band diagram of the source–PTS–drain in NFETs with T_SD_ = 5 nm (solid line) and T_SD_ = 0 nm (dashed line). The E_c_ of the PTS region with different T_IS,B_ at (**b**) T_SD_ = 0 and (**c**) T_SD_ = 5 nm.

**Figure 6 nanomaterials-12-03349-f006:**
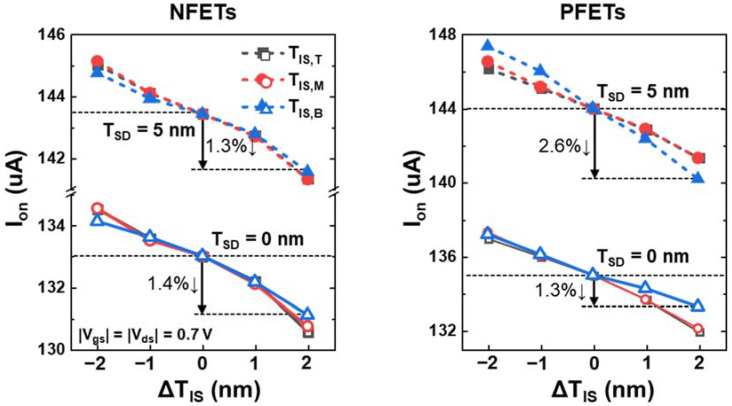
I_on_ of NSFETs having different ΔT_IS_ with T_SD_ = 5 nm (solid symbols) and T_SD_ = 0 nm (open symbols).

**Figure 7 nanomaterials-12-03349-f007:**
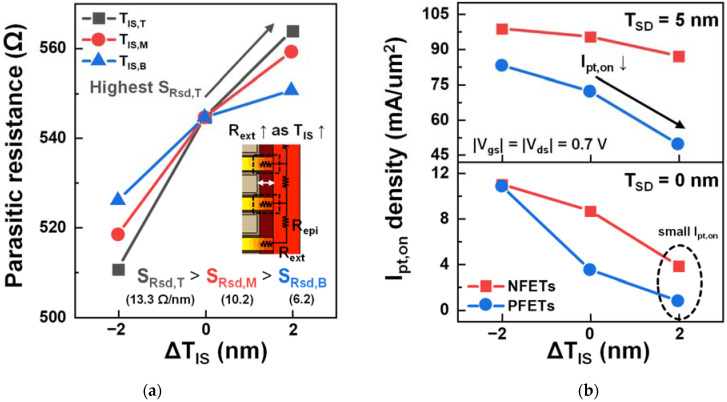
(**a**) Parasitic resistance (R_sd_) of NFETs with respect to the ΔT_IS_. (**b**) I_pt,on_ density of NSFETs with respect to the ΔT_IS,B_.

**Figure 8 nanomaterials-12-03349-f008:**
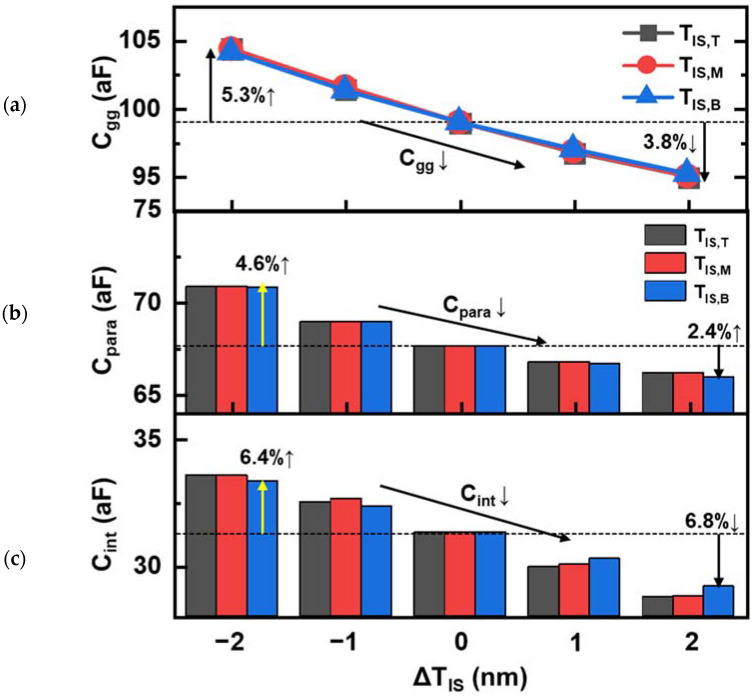
(**a**) C_gg_, (**b**) C_para_, and (**c**) C_int_ for NFETs with respect to ΔT_IS_ (T_SD_ = 0). The capacitances were extracted at a frequency of 1 MHz.

**Table 1 nanomaterials-12-03349-t001:** Geometric parameters for sub-3-nm node NSFETs.

Fixed Parameters	Values
Contact poly pitch (CPP)	42 nm
Fin pitch (FP)	60 nm
Gate length (L_G_)	12 nm
Spacing thickness (T_SP_)	10 nm
NS thickness (T_CH_)	5 nm
NS width (W_NS_)	25 nm
Interfacial layer thickness (T_IL_)	0.6 nm
HfO_2_ thickness (T_HK_)	1.1 nm
T_IS_ without variation (T_IS,ref_)	5 nm
S/D doping concentration (N_SD_)	4 × 10^20^ cm^−3^
PTS doping concentration (N_PTS_)	3 × 10^18^ cm^−3^
**Variable parameters**	**Values**
Excess S/D recess depth (T_SD_)	0 or 5 nm
Inner spacer thickness (T_IS_)	3–7 nm

**Table 2 nanomaterials-12-03349-t002:** DIBL of NSFETs according to the T_IS,B_ and T_SD_.

Type	T_IS,B_ [nm]	DIBL [mV/V]	
		T_SD_ = 0 nm	T_SD_ = 5 nm
NFETs	3	60	67
	5	62	72
	7	67	81
PFETs	3	51	54
	5	53	57
	7	58	61

## Data Availability

Not applicable.

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
