# Peer review of "Sensitivity of Inner Spacer Thickness Variations for Sub-3-nm Node Silicon Nanosheet Field-Effect Transistors"

_nanomaterials, 2022, doi:10.3390/nano12193349_

Round 1

Reviewer 1 Report

The paper describes the (simulated) effects of variations in the thickness of the inner space thickness in combination with variations in the source and drain depths. The paper focusses on the effects on the DC response of the devices. While it is claimed in the abstract that the AC response has also been investigated, this is very minimal and only covered by a more speculative description.

While it is clear that the authors vary the thickness of the inner spacer layers, it has not been made clear what the physical effect of this is. In a device it is unlikely that there will be an empty space, in other words a variation of the Tis would replace some of the semiconductor material. However, the schematic in the paper appear to indicate that this is replace with a vacuum. It would stand to reason that these two have a completely different response and as such it should be made clear.

Some minor issues:

        - The abstract is a representation of the whole paper, as such I would have expected to see a sentence that highlights the main conclusions drawn from this study. It currently concludes with a sentence stating that a comparison is made.

-          - Figure 1, there is no mention of what the colors represent in this figure.

-          - Figure 4b. it is not stated what the Tsd is in this figure.

-          - Figure 5. The graph could be improved by showing the boundaries of the nanosheets.

Reviewer 2 Report

The paper investigates the variations of the inner spacer thickness in sub-3-nm node 3-stacked nanosheet field-effect transistors by using TCAD simulation. It compares the sensitivities of the electrical characteristics that are influenced by the top, middle, and bottom inner spacer. Also, they discuss the effect of the parasitic bottom transistor by considering the S/D recess depth. This paper is novel and interesting. The overall quality is good. I would suggest revising it for publication after a major revision of the following requirement.

1.      In this paper, the simulation results are interesting. However, is there any calibration or other methods that can provide the validation of the simulation results?

2.      In Figure 5, the off-state barrier height of the source-PTS-drain would be lowered as TIS,B or TSD increases. Does the DIBL also be enhanced under such situations?

3.      How did the author extract the on-state current and the off-state current in your discussion?

Round 2

Reviewer 2 Report

I have no more question.